# Molecular Mechanisms of Drug Resistance in Glioblastoma

**DOI:** 10.3390/ijms22126385

**Published:** 2021-06-15

**Authors:** Maya A. Dymova, Elena V. Kuligina, Vladimir A. Richter

**Affiliations:** The Institute of Chemical Biology and Fundamental Medicine, Siberian Branch of the Russian Academy of Sciences, 630090 Novosibirsk, Russia; kuligina@niboch.nsc.ru (E.V.K.); richter@niboch.nsc.ru (V.A.R.)

**Keywords:** glioblastoma multiforme, drug resistance, molecular mechanism

## Abstract

Glioblastoma multiforme (GBM) is the most common and fatal primary brain tumor, is highly resistant to conventional radiation and chemotherapy, and is not amenable to effective surgical resection. The present review summarizes recent advances in our understanding of the molecular mechanisms of therapeutic resistance of GBM to already known drugs, the molecular characteristics of glioblastoma cells, and the barriers in the brain that underlie drug resistance. We also discuss the progress that has been made in the development of new targeted drugs for glioblastoma, as well as advances in drug delivery across the blood–brain barrier (BBB) and blood–brain tumor barrier (BBTB).

## 1. Introduction

Glioblastoma multiforme (GBM) is the most common and devastating primary brain tumor, is highly resistant to conventional radiation and chemotherapy, and does not lend itself to effective surgical resection. According to the data of the National Cancer Institute, about 23,890 adults were diagnosed with new cases of brain and other nervous system cancer in 2020. It also estimates that in 2020, 18,020 of these diagnoses resulted in death [1]. The incidence of GBM ranges from two to three per 100,000 adults per year, and this tumor for 52% of all primary brain tumors. About 17% of all tumors of the brain (primary and secondary) are GBM and develop mostly in people from 45 and 70 years of age. The median survival rate is only 15 months after the first diagnosis and with standard surgery followed by concurrent radiotherapy and chemotherapy, despite advances in surgical and medical neuro-oncology [2].

Methods for diagnosing glioblastoma include both neurological examination and imaging tests (magnetic resonance imaging, positron emission tomography, computerized tomography) [3]. Often, it is necessary to carry out a differential diagnosis of GBM from intracranial mass lesions; here, collection and analysis of tumor tissue via brain biopsy or surgical resection can come to the rescue. The symptoms of glioblastoma can be obscured by other diseases of the central nervous system, which also complicates the diagnosis. Unfortunately, despite the treatment, about 70% of these tumors recur with de novo or acquired resistance, which leads to a low 5-year survival rate [4,5]. Therefore, it is necessary to develop the diagnosis and treatment of GBM using different imaging approaches and targeted drugs.

GBM develops from astrocytes and spreads to nearby brain tissue [6]. Some restructuring of the classification of tumors of the Central Nervous System was made by the World Health Organization in 2016 using molecular and histological parameters [7]. GBM was divided in: (1) isocitrate dehydrogenase (IDH)-wild type (about 90% of cases), which consists of giant cell glioblastoma, gliosarcoma, epithelioid glioblastoma; (2) IDH-mutant glioblastoma (about 10% of cases); (3) glioblastoma, NOS, for which full IDH evaluation is impossible. GBM is one of the best characterized genomic cancers, several types are distinguished according to their transcriptional profile (proneural, neural, classical, and mesenchymal or edge, edge-like, core, core-like), genetics (mutations in IDH gene), and epigenetics (CpG island methylation phenotype (CIMP), O^6^ methylguanine-DNA methyltransferase (MGMT) promoter methylation) [8,9,10,11,12].

## 2. The Diffuse Infiltrative Growth of GBM

The main features of GBM are a diffuse infiltrative growth and an aggressive behavior in combination with a high degree of drug resistance and recurrence. The first feature (diffuse infiltrative growth) affects the formation of drug resistance [13]. Tumor cells extend extra-long membrane projections and use these individual tumor microtubes (TMs) as pathways for invasion, proliferation, and connection to the brain. Thus, functional multicellular network structures are formed. TMs are used for recovery when the network is damaged. In addition, contributing to the development of neoplastic processes, the newly formed vasculature contributes to the expansion of the radius of the tumor spread. Particularly, it was shown that TM-connected glioma cells were more resistant to the cytotoxic effects of temozolomide. Growth-associated protein-43 (GAP-43) was found to be critical for TM formation [14]. Recently, using 2D and 3D models, it was shown that GSLCs formed functional TMs which allowed mitochondria transfer [15]. In the next section, we will consider the main issues related to the formation of drug resistance in GBM.

## 3. Barriers in Brain and Advances in Drug Delivery across the Blood–Brain Barrier

A huge challenge in chemotherapy of brain tumors is the presence of barriers—the blood–brain barrier (BBB) and the blood–brain tumor barrier (BBTB) [16]. These selectively permeable protective barriers reduce the effectiveness of anticancer therapy for human GBM [17]. It has been shown that 98% of low-molecular-weight drugs and almost all therapeutic agents with large molecular weight such as recombinant proteins and peptides, antibodies, and viral vectors, including adeno-associated viruses, do not cross the BBB [17,18]. Endothelial cells of microvessels and capillaries of the brain, connected by tight contacts, forming the BBB, limit the intercellular transport of hydrophilic drugs with a molecular weight > 500 Da. On the other hand, many lipophilic synthetic molecules are limited in access to the brain due to the presence of polarized efflux transporters [19]. For example, only ~0.1% of intravenously administered therapeutic antibodies reach the brain, and high circulating concentrations or an additional administration are required. However, in the later stages of brain tumors, there is a breakdown of the BBB and cerebral edema. It has been shown that high-grade astrocytomas secrete vascular endothelial growth factor, which stimulates angiogenesis, decreases occludin regulation (claudin-1, claudin-5), redistributes astrocyte AQP4, and increases endothelial cell permeability [20,21,22,23].

The attempts to increase therapeutic concentrations at their target levels with an intact BBB may lead to an increased risk of systemic toxicity [24]. To overcome the existing barriers and increase the effectiveness of therapy for brain diseases, various invasive and non-invasive approaches have been developed, which, in turn, have both advantages and disadvantages. The invasive approaches include intrathecal administration by the direct administration or catheters [25], convention-enhanced delivery via bulk flow [26], interstitial wafers, and implants [27]. The non-invasive approaches suggest the usage of chemical modification of drugs such as lipidization [28], substrates for carrier-mediated transcytosis [29], substrates for receptor-mediated transcytosis, virus-mediated blood–brain barrier delivery [30,31], and exosome-mediated blood–brain barrier delivery [32]. There are also several methods different from the above two approaches: the intranasal delivery route [33], the modulation of the blood–brain barrier permeability by hyperosmotic agents [34], and the use of focused ultrasound [35].

With regard to the development of cancer drugs for selective delivery to cancer cells of glioblastoma, a modular principle was proposed [17,36]. The first module or component is a carrier that facilitates tumor targeting. Small molecules and biological substances such as peptides, aptamers, proteins, and antibodies are well suited for this role, which allows sufficient tumor selectivity. The second component is an oncotoxic drug, such as a cytotoxic molecule or radionuclide. The third component is the link between the first and the second components. Examples of successful BBB penetration include the blood–brain barrier (BBB)-penetrating peptide angiopep-2 loaded with three PTX molecules by ester linkers (ANG1005) [37], clusters of sodium borocaptate (BSH) conjugated with the 11R peptide for boron neutron capture therapy [38], an arginine-containing tripeptide modified with BSH and a DOTA chelator with significantly high tumor-to-normal-brain and tumor-to-blood radioisotope accumulation ratios [39].

## 4. Molecular Features of Glioblastoma Cells Which Promote Chemoresistance

As other tumors, GBM has a huge number of different pathways for the emergence of drug resistance due to its heterogeneity, hypermutation, the Warburg effect, immune evasion, oncologically activated alternative splicing pathways, etc. [40,41].

### 4.1. Heterogeneity

One of the characteristics of glioblastoma that interferes with effective chemotherapy and radiotherapy is its heterogeneity: interpatient, intratumoral, functional, and molecular heterogeneity [41,42]. As in the case of other cancers, the heterogeneity of glioblastoma can be explained by both clonal evolution and the presence of stem cells.

According to the clonal evolution theory, successive mutations accumulated in a cell give rise to clonal outgrowths that proliferate in response to tumor microenvironment selective pressures such as hypoxia, acidosis, competition for space and resources, immune evasion, and treatments. It is believed that, at first, a benign neoplasia forms, and only with the accumulation of mutations does malignant transformation occur. Hereditary changes in the epigenome may also impart beneficial traits to variant subpopulations, facilitating their clonal expansion. It is noted that in the late stages of tumor progression, the creation of clonal outgrowths occurs faster, which may be the cause of significant interclonal heterogeneity [43].

Inside the tumor, a pool of cells capable of differentiation and increased proliferation, the so-called cancer stem cells (CSCs), is isolated. These cells are at the top of a hierarchy of progressively differentiating cells and are responsible for the initiation and maintenance of the tumor after treatment. Recently, using massively parallel single-cell RNA sequencing, a conservative trilineage cancer hierarchy was discovered with a population of progenitors at the apex, composed of cyclic cancer cells and functionally resembling GSCs [44]. By RNA sequencing, it was shown that glioblastoma presents a normal neurodevelopmental hierarchy. Many molecular mechanisms have been associated with the resistance of CSCs to cytotoxic therapies, including mechanisms involving the G_2_-M DNA damage checkpoint, different signaling pathways (Notch, NF-κB, EZH2, and PARP), Wnt/β-catenin signaling cascade [45,46,47,48,49]

Moreover, different subtypes of GBM with distinct molecular profiles coexist within the same tumor and likely exhibit differential therapeutic responses [50]. A comparison of tumor tissue and isolated stem-like subpopulations revealed 418 genes upregulated in tumor tissue with respect to CD133-positive/CD15-positive cells and 44 genes upregulated in CD133-positive/CD15-positve cells with respect to tumor tissue. These upregulated genes are relevant to the cell division cycle and oncogenesis [51]. In general, receptor tyrosine kinases (RTKs) are dysregulated in approximately 90% of GBM, and often platelet-derived growth factor receptor (PDGFR) and epidermal growth factor receptor (EGFR) are amplified or mutated in GBM [52,53]. Abnormal RTK activation involves multiple pathways that can initiate and maintain downstream signaling and thus may play a critical role in therapeutic resistance and glioblastoma recurrence [54]. Furthermore, an increase in tumor heterogeneity was associated with a decrease in patient survival [40,55]. The different mechanisms of drug resistance may also require the use of a combination of targeting agents. Moreover, it was shown that cells of the same tumor have a differential expression of genes involved in oncogenic signaling, proliferation, immune response, and hypoxia. The radiation resistance of glioma may be also associated with an increase in the DNA damage response in CSCs [45].

It has recently been found that multiple long non-coding RNAs (lncRNAs) are involved in a wide range of biological functions, including angiogenesis, through the regulation of gene expression in cancer [56]. SLC26A4-AS1 lncRNA has been reported to promote NPTX1 transcriptional activity by recruiting nuclear factor kappa B 1 (NF-κB 1), which has an anti-angiogenic effect on glioma cells; the problem is that SLC26A4-AS1 expression is suppressed in glioma cells. Hypoxic glioma stem cells (H-GSCs) secrete exosomes containing lncRNAs (Linc01060), which in turn activate prooncogenic signaling pathways, clinically promoting disease progression [57].

### 4.2. Hypermutation

Hypermutation is common in tumors that recur after the use of alkylating agents (post-treatment hypermutation) and conversely, rarely occurs in newly diagnosed gliomas (de novo hypermutation) [58]. It is assumed that there are two main reasons of hypermutation: a de novo pathway connected to constitutional defects in DNA polymerase and a pathway involving mismatch repair (MMR) genes. The latter is more commonly activated after treatment and is associated with acquired resistance driven by MMR defects in chemotherapy-sensitive gliomas that recur after treatment with temozolomide. Alternative pathways for the emergence of a hypermutation status due to stem cells are also possible. Therefore, when exposed to TMZ chemotherapy, as a result of greater drug efflux activity and slower proliferation rate, stem cells can cause both a new non-hypermutant and a hypermutant recurrence [59]. The emergence of hypermutation is also associated with the simultaneous use of radiation together with TMZ for the treatment of glioma. Radiation can induce the expression of O^6^ methylguanine-DNA methyltransferase (MGMT) [60] and can also induce a temporary growth arrest, which can provide temporary resistance to TMZ-induced mutagenesis [61]. Whether the presence of hypermutation in GBM is associated with a better or worse outcome for patients remains an open question. So far, mouse models of genomic instability have shown a decrease in tumor growth with an increasing mutational burden [62]. Therefore, an increase in the number of mutations in hypermutant cells after radiotherapy can lead to a decrease in fitness, making the cells less aggressive and more amenable to additional treatment.

### 4.3. The Warburg Effect

The Warburg effect is also observed in GBM [63]. It is characterized by an increase in the rate of glucose uptake and a predominant production of lactate even in the presence of oxygen. Thus, aerobic glycolysis is used to produce ATP for the rapid proliferation of cells with high bioenergetic and biosynthetic needs. Glucose, used in glycolysis and oxidative phosphorylation of mitochondria, is also required for biosynthetic pathways such as those involving ribose sugars for nucleotides, nonessential amino acids, glycerol and citrate for lipids, and NADPH via the oxidative pentose phosphate pathway. To meet the biosynthetic needs of proliferating cancer cells, mitochondria, which serve as the main bioenergy center, also provide metabolites for the synthesis of macromolecules. Moreover, the distinct molecular characteristics of glioblastoma stem cells (GBM18, GBM27, GBM38), as well as U87MG, influence the metabolic phenotypes even under the same cell culture conditions. A more thorough characterization of GBM, both glycolytic and oxidative subpopulations, is required for a more effective metabolic therapy. Due to high glucose uptake, it could be possible to used 2-^18^F-deoxyglucose with PET as a means of diagnosing and monitoring cancer treatment response. Unfortunately, standard procedures such as PET with 2-^18^F-deoxyglucose do not differentiate between high glucose uptake due to increased aerobic glycolysis or oxidative phosphorylation, or low glucose uptake due to compensatory glutaminolysis and necrosis [64]. Different enzymes which are involved in aerobic glycolysis can be used as a target for cancer treatment, for example, fasentin, ritonavir, WZB117, STF-31, as inhibitors of the glucose transporter GLUT1; 3-BrPA, 2-DG, lonidamine as inhibitors of the hexokinase; PFK15, 3PO as inhibitors of 6-phosphofructo-2-kinase; OA, TT-232, VK3, VK5 as inhibitors of the M2 isoform of pyruvate kinase; rapamycin as an inhibitor of mTORC1; DCA as a PDK1 inhibitor; NAC as an antioxidant, preventing oxidative stress; hydroxy-chloroquine as an autophagy inhibitor; metformin as an inhibitor of oxidative phosphorylation; and halofuginone as an Akt/mTORC1 inhibitor [65].

### 4.4. Immune Evasion

Tumor immune evasion is a serious and complex problem that can lead to failure of GBM treatment. Glioblastoma is located in the brain, which was previously considered an immuno-privileged organ, but now, with the emergence of new data, it is believed that the brain is an actively regulated site of immune surveillance [66]. Yet, if the blood–brain barrier is not violated, the infiltration of peripheral immune cells into a tumor is limited. If the BBB is disrupted by tumor proliferation or inflammation, then the tumor can be infiltrated by immunosuppressive immune cells [67]. It has been shown that an immunosuppressive microenvironment in a glioma is created by immunosuppressive cytokines and chemokines (transforming growth factor beta (TGF-β), interleukin 10 (IL-10), prostaglandin E2) and immune cells (immunosuppressive natural killer T cells (NKT), T/B regulatory cells (T/Breg), tumor-associated macrophages (TAM)/microglia and myeloid-derived suppressor cells (MDSC)). It has a pro-oncogenic activity that leads to tumor progression. In these conditions, the expression of immune checkpoint receptors (cytotoxic T lymphocyte-associated antigen-4 (CTLA-4) and programmed cell death protein-1 (PD-1)) is dramatically increased. These immune checkpoint receptors, that are expressed on T cell surface, play a negative regulatory role during the process of T cell activation, thereby preventing pathological overactivation [68]. However, glioblastoma therapy with immune checkpoint inhibitors has not been successful due to resistance to PD-1 or CTLA-4 blocking antibodies. At the moment, a search is underway for new inhibitors of immune checkpoints for their use both in monotherapy and in combination with other drugs.

Tumor-associated macrophages (TAMs) are the dominant immune population in glioma, constituting around 30–40% of the total tumor volume. The tumor microenvironment (TME) attracts TAMs to the tumor and polarizes them to an anti-inflammatory or pro-tumorigenic “M20-like state”. Because of this, current studies are aimed at inhibiting TAM recruitment or survival in the TME, allowing for their functional reeducation to an anti-tumor “M10-like state” or by targeting the tumor using monoclonal antibodies that elicit macrophage-mediated phagocytosis and intracellular destruction of cancer cells.

Spontaneously occurring tumors contain weak tumor antigens. Moderate immune responses to a tumor or their complete absence are explained by the absence of expression of foreign genes in tumor cells. In addition, if the tumor mutation burden, the level of DNA mismatch repair, and genomic microsatellite instability are low, the production of tumor antigens will be reduced, which may lead to drug resistance [69]. This can also cause anatomical isolation of tumor antigens from the immune system (in particular, due to the additional barriers for brain tumors—BBB and BBTB) or generalized immunosuppression in a cancer patient. The latter is typical for many disseminated tumors at late stages of development.

Another reason for immune evasion is the lack of recognition of tumor cells by T lymphocytes due to inadequate or low presentation of antigens. This is often due to the fact that cancer cells often lose class I and/or II histocompatibility antigens required for antigen presentation [70]. To stimulate the differentiation of precursor T cells to cytotoxic T lymphocytes, it may be necessary to express not only antigens for binding to T cell receptors by tumor cells, but also ligands to ensure the adhesion of T cells to tumor cells for their co-stimulation [71]. One of the explanations for the non-recognition of tumor antigens by T lymphocytes is precisely the insufficient expression of these ligands, in particular ICAM1, LFA1, VCAM-1, B7 family members [72]. Recently, it was shown that tumor-associated immune evasion is due to the downregulation of TLR4 expression in glioblastoma and TAM cells. Moreover, this was more pronounced in resistant glioblastoma cells, with more resistant GBM showing lower TLR expression compared to sensitive cells [73].

In addition, tumor cells can be resistant to the action of cytotoxic cytokines secreted by effector cells, or even be stimulated rather than suppressed by these cells [74,75]. Cellular immunity may not be able to block tumor growth also because tumor cells multiply too quickly or their number is initially too large.

Returning to the topic of CSCs, it should be noted that there is evidence that CSCs directly modulate the immune system. In co-culture systems, CSCs induced regulatory T cells while inhibiting the proliferation and activation of cytotoxic T cells, with a concomitant induction of cytotoxic T cell apoptosis, mediated by PD1 and soluble galectin-3 [76,77]. Other CSC-secreted factors include IL-10 and TGF-β, which also suppress tumor-associated microglia/macrophage function and generate a more immunosuppressive (M2) phenotype [78]. Another immunotherapy approach that may benefit from CSC targeting is the development of anti-tumor vaccines. Current vaccine efforts have focused on tumor-specific antigens (such as EGFRvIII) or whole tumor cell lysates, and there is evidence from preclinical models that CSC lysates are more effective in generating dendritic cell (DC) vaccines than differentiated cells [79,80]. CSCs modulate T cell and tumor-associated microglia/macrophage function through secreted factors [81], which may be exploited in the development of vaccine strategies or of combinations with other drugs [82]. These data provide a rationale for future studies investigating how the interaction between CSCs and other immune cell populations may drive immune suppression and stimulate in vivo investigations into how CSC targeting may alter the immune activation status. Evaluating changes in CSC populations, as a result of immunotherapy, will also be essential, as will be evaluating combinatorial targeting strategies using immunotherapies and anti-CSC approaches.

### 4.5. Oncologically Activated Alternative Splicing Pathways

Alternative splicing is the source of a variety of transcriptional variants and regulatory elements [83]. A large number of mutations and alterations in splicing factors have been shown in tumors; thus splicing can be considered a cancer driver. Changes in RNA processing, contributing to the acquisition of cancer cell phenotypes can be due to changes in the levels of expression and activity of RNA-binding proteins, RNA-processing factors, mutations in splice sites, or regulatory elements [84].

Among the regulators of RNA processing, several RNA-binding proteins are distinguished, which are involved in the development of cancer. The small nuclear ribonucleoprotein-associated proteins B (SNRPB), which are the main component of the spliceosome, have been identified as the main effectors of cell viability, proliferation, and apoptosis. Knockdown of the SNRPB gene affected the splicing of genes involved in RNA processing, DNA repair, and chromatin modulation [83]. The polypyrimidine tract-binding protein PTB also plays a role in RNA processing and cancer development. It is highly expressed in different brain tumors (low-grade astrocytoma, anaplastic astrocytoma, medulloblastoma, paraganglioma, GBM, etc.). It has also been shown that the development of GBM involves proteins such as heterogeneous nuclear ribonucleoproteins, hnRNPH, human antigen R (HuR), RBM14, etc. [85]. RNA processing in tumor cells may change due to mutations in the genes of splicing regulators (SF3B1, SRSF1, RBM4, RBM5/6/10, U2AF1, clk/STY, and CLK2). RNA processing studies have shown that cancer can be uniquely susceptible to splicing modulation therapy, in contrast to normal body tissues. An example is the study of the effect of the loss of the SF3b complex protein PHF5A or its drug inhibition, which caused massive cell death only in cancer cells. Several drugs that are involved in RNA binding and splicing pathways have been tested and show promising results for the treatment of GBM (metformin, amiloride, pladienolide B, AR-A 014418).

### 4.6. Role of microRNAs in the Resistance of Glioblastoma

MicroRNAs are a class of small noncoding RNAs which are involved in the regulation of a variety of cellular processes by degrading their target mRNAs and/or inhibiting their translation in both normal and disease physiological conditions. Therefore, miRNAs could be influential on the initiation, progression, and metastasis of glioblastoma multiforme (GBM). The level of several miRNAs was evaluated in the context of the development and further prognosis of GBM. For example, the level of miR-1271 that targets Bcl-2 was decreased in samples from patients with GBM [86]. It was also found that miR-133a suppressed DR5 expression and activated NF-κB signaling, thus contributing to the resistance to tumor necrosis factor-related apoptosis-inducing ligand (TRAIL) [87]. miR-519a targets the STAT3/Bcl2 signaling pathway, enhanced chemosensitivity, and promoted autophagy [88]. The interaction of the microRNAs (miRNAs) with RBPs may be advantageous for the development of new drugs. It has been shown that miR-10b is a promising candidate for the development of new therapies against GBM, since the modulation of its function affects the cell cycle and the regulation of splicing in CSCs and reduces the growth of intracranial GBM xenografts. Using an oncogenic RNA splicing machinery, which induces the simultaneous alteration of a vast numbers of transcripts and proteins, to develop drugs against GBM may reduce the chance of therapeutic escape due to heterogeneity or adaptation [83]. Recently, it has been announced that the new mechanism of post-transcriptional gene regulation, known as the certain competing endogenous RNA regulatory networks (ceRNETs), controls the expression of related genes by microRNA sponges. Due to the poor prognosis of GBM and the short average survival time of patients, researches in this area is promising for the development of early diagnosis of the disease. In general, judging by the huge number of publications over the past five years, we hope that this will bring new discoveries and prospects for the development of new diagnostic systems and targeted drugs.

## 5. Current Drugs for the Treatment of GBM and Possible Mechanisms of Resistance to Them

### 5.1. Temozolomide (TMZ)

Currently, one of the main chemotherapeutic drugs for GBM is temozolomide (TMZ), which is a pro-drug, that methylates DNA at the O^6^ position of guanine [89]. These methyl adducts lead to a mismatch pairing of guanine with thymine rather than cytosine during the S phase and DNA replication [90]. This leads to genomic instability and eventually cell death [59]. The increased expression of *O-*6-methylguanine-DNA methyltransferase (MGMT) drives resistance to TMZ [91]. Therefore, the methylation status of the MGMT promoter could be a potential marker of the response to TMZ treatment. CSCs that were CD133-positive and had high expression of breast cancer resistance protein1 (BCRP1) and MGMT showed resistance to TMZ [92]. Using transcriptome microarray and bioinformatic analyses, it was found that TMZ reduced Notch3 levels, which are upregulated in gliomas, by inducing CHAC1. CHAC1binds to the Notch3 protein and inhibits Notch3 activation, thus CHAC1-inhibited Notch3 signaling can influence TMZ-mediated cytotoxicity [93].

Activation of phosphoinositide 3-kinase (PI3K) signaling leads to resistance of GBM to alkylating drugs [94]. It was shown that small noncoding RNAs such as miRNAs can participate in the resistance to TMZ. For example, by targeting Bcl-2, miR-181b-5p inhibits proliferation, migration, and invasion of the glioma cell line U-87 MG and also enhances the tumor-suppressive effect of TMZ [95]. In turn, miR-181d could be a potential marker of TMZ resistance in GBM, because its levels are negatively associated with MGMT gene expression [96]. Overexpression of miR-132 leads to downregulation of tumor suppressor candidate 3 (TUSC3), which inhibits the formation of CSCs. Thus, the overexpression of miR-132 induces temozolomide resistance and promotes the formation of CSCs phenotypes [97]. Using a gene set enrichment analysis (GSEA), it was shown that epithelial-to-mesenchymal transition (EMT) is associated with poor TMZ responses. In addition, miR-140 targeting cathepsin B (CTSB) signaling inhibits the mesenchymal transition and enhances temozolomide cytotoxicity in GBM [98].

Recently, it was shown that overexpression of FAM289 contributes to tumor progression in glioma cell lines. The interaction of FAM289 with galectin-1 facilitates its entry into the nucleus where FAM289 activates the extracellular signal-regulated kinase (ERK) pathway, upregulates DNA methyl transferase 1 (DNMT1) expression, and induces the formation of the CSCs phenotype, which leads to drug resistance of glioma cells to TMZ [99]. Analysis of the transcriptome of GBM hypoxic cells showed that the Jun proto-oncogene (*JUN*), transcriptional factors CCAAT/enhancer binding protein beta (CEBPB), and histone deacetylase 3 (HDAC3) were closely involved in the drug-resistant phenotype of hypoxic GBM [100]. Whereas the activation of the transcription factors JUN and CEBPB is associated with tumorigenesis, the suppression cell cycle events are connected with HDAC3. Thus, the proteins FAM289, JUN, CEBPB, and HDAC3 can be seen as potential therapeutic targets for cancer treatment. It is worth noting that the transcription factor CCAAT/enhancer-binding protein α (CEBPα), which is involved in differentiation of myeloid cells, proliferation, metabolism, and immunity, has been already considered for the development of MTL-CEBPA, the first-in-class small activating RNA oligonucleotide drug [101]. It was shown that honokiol improved the effects of TMZ on autophagy and apoptosis of GBM cells [102]. Therefore, an understanding of the mechanisms of intrinsic resistance to TMZ may facilitate the choice of a strategy for the development of new anticancer drugs or adjuvant drugs to enhance therapeutic responses.

### 5.2. Other Drugs

EGFR kinase inhibitors (erlotinib, gefitinib) are used in the therapy of GBM, but only 10–20% of patients respond to them. As mentioned above, EGFR is frequently amplified, overexpressed, or mutated in glioblastomas [103]. It was suggested that mutations in the EGFR gene do not always correlate with resistance to EGFR kinase inhibitors [104]. The constitutively active genomic deletion variant of EGFR, EGFRvIII, is often expressed in GBM. In turn, EGFRvIII persistently activates the PI3K signaling pathway and may sensitize GBM to EGFR kinase inhibitors. The inhibitor of the PI3K signaling pathway phosphatase and tensin homologue deleted in chromosome 10 (PTEN) is often lost in GBM. It was shown that EGFRvIII may sensitize GBM to EGFR kinase inhibitors, whereas the lack of PTEN is associated with resistance to them. It was also assumed that PTEN may influence drug resistance through by regulating the expression of quinone oxidoreductase 1 (NQO1) and of a PTEN-induced putative kinase 1 (PINK1) [103].

For the treatment of glioblastomas, various therapeutic approaches can be used, but all have certain limitations. The fatty acid oxidation (FAO) inhibitors Perhexiline could be a promising therapeutic drug due to its central role in GBM metabolism [105]. Perhexiline induced potent redox stress and apoptosis in vitro in GMM cells. Perhexiline sensitivity or resistance depends on the expression of FYN, a mediator of perhexiline-induced cytotoxic. The pan-adenosine triphosphate-binding cassette transporter and L-type voltage-dependent calcium channel inhibitor verapamil with carmustine and irradiation, made the glioma cell line U-87 MG and patient-derived CSCs more sensitive to therapy-induced senescence through MAPK, phosphoinositide 3-kinase (PI3K)/AKT, and NF-κB signaling pathways [106]. Unfortunately, this calcium channel blocker provokes strong cardiotoxicity at a dose 10-fold lower than that cytotoxic to GBM cells [107].

Virus-like particles (VLPs) such as the recombinant poliovirus/rhinovirus chimera (PVSRIPO) have proven to be an effective and useful candidate for the treatment of glioblastoma [108]. The attenuated Zika virus (ZIKV) vaccine candidate (ZIKV-LAV) efficiently binds to glioma CSCs in vivo [109]. Oncolytic viruses can be used both as a monotherapy and in combination with other traditional methods and treatment regimens for GBM [110]. It is worth noting that the response to viral therapy depends on an innate antiviral response, which is mainly orchestrated by Type-I interferons (IFN-I). IFN-I limits viral spread through the IFNAR–JAK–STAT pathway. IFN-I participates in the development of antigen-specific adaptive immune responses, antigen presentation, and natural killer (NK) cell function. However, in the case of glioblastoma, the action of IFN-Is seems to be contradictory due to the contribution of autocrine IFN-I signaling to immune evasion of glioma cells, constitutive activation of STAT proteins, and negative regulation of antigen-presenting capacity [111]. Since GBM primarily affects older people and chronically elevated levels of IFN-I in the brain are associated with aging, it is suggested that tumors have evolved and adapted to carry or even exploit active IFN-I signaling to their advantage [112].

There are several potential drugs that showed promising results in in vitro or in vivo experiments, but their molecular mechanisms of action have not yet been studied, and it is too early to make promising predictions. It was shown that SP-141 was cytotoxic for GBM cells because of decreasing MDM2, increasing levels of p53 and p21cip1, arrest of the G2/M cell cycle, and severe apoptosis [113]. Eupatiline suppressed the viability and proliferation of glioma cells, reduced migration and invasion, and suppressed tumor growth in vivo, but did not promote apoptosis [114]. The curaxin CBL0137, by inactivating the chromatin remodeling complex, activated p53 and also downregulated nuclear factor-kappa B (NF-ĸB) and Facilitates Chromatin Transcription (FACT) [115]. By generation of ROS in the glioma cell line U-87 M, salinomycin caused endoplasmic reticulum stress-mediated autophagy and apoptosis [116]. The exercise-induced myokine irisine suppressed tumor proliferation, invasion, and growth of GBM cells by G_2_/M cell [3] cycle arrest and by increasing the levels of p21 [117]. It was shown that sulforaphane (SFN), which is obtained from cruciferous vegetables, induced apoptosis in tumor tissues. This molecules has been modified to sulforaphane-cysteine (SFN-Cys) in order to increase its half-life and thus its enrichment in plasma [118]. The induction of cell apoptosis by this SFN-Cys involves the long-term activation of ERK1/2- and ERK1/2-mediated signaling pathways such as the activation of caspase 3 and proteins associated with apoptosis. Macromolecules such as DNA- or RNA-aptamers [3,119], antibodies [120], metal core nanoparticles (NPs), peptide nanoparticles [121,122,123], liposomes (ImmuLipCP) [124], CAR T cells [125], and CRISPR/Cas tools [126] are gradually included in a number of therapeutic drugs or theranostics for glioblastoma in in vivo or in vitro experiments.

## 6. Conclusions

Resistance to conventional therapies and frequent recurrence are major barriers to the treatment of glioblastoma. Thus, the development of new therapeutic strategies is imperative to overcome these barriers and improve treatment outcomes. In this review, an attempt was made to highlight the main issues related to the development of drug resistance in glioblastoma (Figure 1).

Resistance can be associated with internal factors, such as the blood–brain barrier (BBB), the blood–brain tumor barrier (BBTB), genetic molecular characteristics (heterogeneity, the Warburg effect, oncologically activated alternative splicing pathways), and external factors in response to a therapeutic agent or the immune system (immune evasion). The role of some factors such as miRNAs and hypermutation, can be controversial. Due to the heterogeneity of GBM, the most promising therapeutic strategy for the treatment of this disease involves combinatorial approaches and treatment regimens. For example, with an intact BBB, drugs that penetrate the BBB (upon chemical modification, use of substrates for carrier-mediated transcytosis and for receptor-mediated transcytosis, virus-mediated and exosome-mediated blood–brain barrier delivery) in combination with hyperosmotic agents and focused ultrasound will be useful. In the case of a destroyed BBB, the choice of a therapeutic agent is wider, but patient prognosis is already poor, since we are talking about patients with stage II–IV cancer. An interesting approach is the modular principle (carrier–link–drug). We tried to elucidate the mechanism of development of GBM resistance to the main therapeutic drug TMZ. Understanding the mechanisms of intrinsic resistance to TMZ may facilitate the choice of a strategy for the development of new anticancer drugs or adjuvant drugs to enhance therapeutic responses. At the moment, a huge amount of knowledge and research has accumulated about the effect of one or another therapeutic agent or approach; Thus, when developing a new molecule, the developer can use the already proposed research algorithms and knowledge to assess the development of drug resistance.

## Figures and Tables

**Figure 1 ijms-22-06385-f001:**
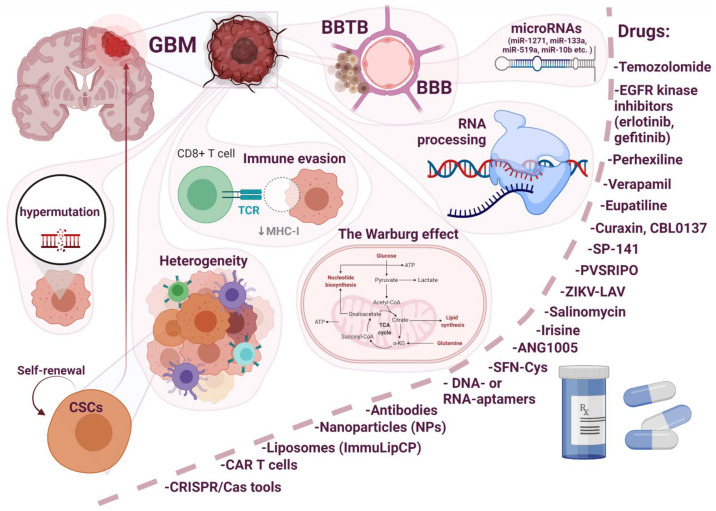
Main issues related to the development of drug resistance in glioblastoma multiforme (GBM).

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
