# Peer review of "Molecular Mechanisms of Drug Resistance in Glioblastoma"

_ijms, 2021, doi:10.3390/ijms22126385_

Round 1

Reviewer 1 Report

The link to the American Association of Neurological Surgeons is no longer available. Instead of citing a web site, the authors should identify peer-reviewed literature that serves their line of argumentation.

GBM is classically of astrocytic origin, there is only few evidence for GBM with an oligodendroglial component that is not very well characterised. Please revise the introduction accordingly. 

Further comments introduction:

Please update your text to the current knowledge of transcriptional subgrouping of GBMs and use the most recent publications for citation. 

For epigenetics, please cite the literature for the G-CIMP phenotype and in this context please also mention MGMT methylation.

Please structure the introduction a bit better, there is certain redundancy on survival information and surgery at the end and beginning of the introduction, so that it reads more fluid.

I would suggest to discuss invasion, infiltrative properties/growth and distant recurrence as a separate chapter and how it contributes to treatment resistance. Please also cite the literature for the microtube theory. 

In the chapter Heterogeneity, only the concept of cancer stem cells is discussed, to provide an more unbiased overview, the authors should discuss and compare the clonal evolution theory and the concept of interclonal cooperativity/tumor cell plasticity. The whole discussion on CSC is matter-of-fact like, which should in my opinion be revised and expanded. Please also discuss the current literature on single-cell experimentation aiming at elucidating GBM tumor heterogeneity. 

After reading the chapter on hypermutation it should also be made clear whether the concepts of hypermutation are discussed in the context of IDHwt GBM or IDHmut high-grade astrocytoma. Please also add references for radiation induced expression of MGMT and TMZ-induced mutagenesis.

Regarding the Warburg effect, please also discuss anti-Warburg therapeutics and biomarkers.

Current drugs for the treatment: redundancy in the wording TMZ is a pro-drug that methylates, please rephrase or adapt. 

Please clarify in the text: are O-6-methylguanine-DNA methyltransferase (MGMT) or O-6-alkylguanine-DNA alkyltransferase 2 different enzymes? What is the abbreviation then for the latter?

Author Response

Author's Reply to the Review Report (Reviewer 1)

Dear Reviewer, thank You very much for a thorough revision of our manuscript. Please find below our response to your comments:

  1. The link to the American Association of Neurological Surgeons is no longer available. Instead of citing a web site, the authors should identify peer-reviewed literature that serves their line of argumentation.

Response: Corrected. We added the reference (https://doi.org/10.1016/j.cell.2021.03.023).

  1. GBM is classically of astrocytic origin, there is only few evidence for GBM with an oligodendroglial component that is not very well characterised. Please revise the introduction accordingly. 

Response: Corrected.

Further comments introduction:

  1. Please update your text to the current knowledge of transcriptional subgrouping of GBMs and use the most recent publications for citation.

Response: Corrected. We decided to add the recent publication (https://doi.org/10.1093/brain/awz044) and the publication from which this division into subgroups began (https://doi.org/10.1016/j.ccr.2009.12.020).

  1. For epigenetics, please cite the literature for the G-CIMP phenotype and in this context please also mention MGMT methylation.

Response: Corrected. We added the references - https://doi.org/10.3389/fonc.2019.01547.

  1. Please structure the introduction a bit better, there is certain redundancy on survival information and surgery at the end and beginning of the introduction, so that it reads more fluid.

Response: We structured the text of the introduction.

  1. I would suggest to discuss invasion, infiltrative properties/growth and distant recurrence as a separate chapter and how it contributes to treatment resistance. Please also cite the literature for the microtube theory.

Response: Corrected. We made the separate chapter for it. We've added the sentences and links to microtube theory.

  1. In the chapter Heterogeneity, only the concept of cancer stem cells is discussed, to provide an more unbiased overview, the authors should discuss and compare the clonal evolution theory and the concept of interclonal cooperativity/tumor cell plasticity. The whole discussion on CSC is matter-of-fact like, which should in my opinion be revised and expanded. Please also discuss the current literature on single-cell experimentation aiming at elucidating GBM tumor heterogeneity. 

Response: Corrected. We added the discussion about the clonal evolution theory, the cell plasticity, the single-cell experimentation.

  1. After reading the chapter on hypermutation it should also be made clear whether the concepts of hypermutation are discussed in the context of IDHwt GBM or IDHmut high-grade astrocytoma. Please also add references for radiation induced expression of MGMT and TMZ-induced mutagenesis.

Response: In this chapter of the review, we'll talk about hypermutation in general, because it's hard to cover everything - we're limited in scope. Particularly, we refer the reader to articles in which, for example, there is specific information. For example, [58]: «Of note, the prevalence of hypermutation in post-temozolomide samples correlated with the chemosensitivity of the primary, molecularly defined tumour type (1p/19q co-deleted oligodendrogliomas (59.5%) > IDH1/2-mutant astrocytomas (30.2%) > MGMT-methylated IDH1/2 wild-type glioblastomas (23.1%) > MGMT-unmethylated IDH1/2 wild-type glioblastomas (5.6%); P = 3.8 × 10−7; Fig. 1b). We observed a similar pattern in the FMI validation dataset (Extended Data Fig. 3g-i)». As we can see there is some difference between IDHwt GBM or IDHmut high-grade astrocytoma. On the other hand, there are many articles where it is written either about the ultramutant IDH-wt GBM (doi: 10.3390 / Cancers11091279), or the hypermutating phenotype driven by TMZ (doi: 10.1016/j.annonc.2020.08.2336), or different polymorphisms (CNV and SNV) in both IDHwt and IDHmut glioblastomas (doi: 10.1155/2019/4878547). This is why we wrote generally without dividing into these two types.

We added the reference for radiation induced expression of MGMT and TMZ-induced mutagenesis.

  1. Regarding the Warburg effect, please also discuss anti-Warburg therapeutics and biomarkers.

Response: We added the discussion of anti-Warburg therapeutics and biomarkers.

  1. Current drugs for the treatment: redundancy in the wording TMZ is a pro-drug that methylates, please rephrase or adapt. 

Response: Corrected.

  1. Please clarify in the text: are O-6-methylguanine-DNA methyltransferase (MGMT) or O-6-alkylguanine-DNA alkyltransferase 2 different enzymes? What is the abbreviation then for the latter?

Response: Corrected the sentence. O-6-alkylguanine-DNA alkyltransferase (AGT) and O-6-methylguanine-DNA methyltransferase (MGMT) are the same enzymes.

All changes in the article are highlighted in yellow to make it easier for the Editors to give a prompt decision on manuscript.

Yours faithfully, Dymova Maya.

Reviewer 2 Report

In this review, the authors discuss a number of mechanism that are related to inefficacy of chemotherapeutic agents in the treatment of glioblastoma. Overall, the article goes through and explains various traditional and also more recently studied causes for drug resistance or inability to appropriately deliver chemotherapeutic drugs.

Some issues need clarification:

In the introduction in line 38 the information is somewhat redundant to line 21. Therefore, it should be deleted.

In line 45 the authors state that only biopy plays a role for definitive diagnosis. This should be stated more comprehensively such as collection and analysis of tumor tissue via brain biopsy or surgical resection.

Following paragraph 3 (listing of the individual molecular features for resistance) it would make reading easier if the following parts would be numbered as 3.1., 3.2. and so forth instead of 4., 5. until the paragraph on current drugs (which is a different topic)

In lines 308 to 310, the information is redundant and should be corrected. Also, in the following sentence, there is the impression that part of the sentence is missing. What did the authors want to state here?

At last, in line 439, the authors state cancer stages 2-4. What exactly do they mean by that (WHO for CNS tumors? This would not be GBM then)? This should be explained a bit more thoroughly.

Author Response

Author's Reply to the Review Report (Reviewer 2)

Dear Reviewer, thank You very much for a thorough revision of our manuscript. Please find below our response to your comments:

In this review, the authors discuss a number of mechanism that are related to inefficacy of chemotherapeutic agents in the treatment of glioblastoma. Overall, the article goes through and explains various traditional and also more recently studied causes for drug resistance or inability to appropriately deliver chemotherapeutic drugs.

Some issues need clarification:

  1. In the introduction in line 38 the information is somewhat redundant to line 21. Therefore, it should be deleted.

Response: Corrected. The introduction was rewritten.  

  1. In line 45 the authors state that only biopy plays a role for definitive diagnosis. This should be stated more comprehensively such as collection and analysis of tumor tissue via brain biopsy or surgical resection.

Response: Corrected.

Following paragraph 3 (listing of the individual molecular features for resistance) it would make reading easier if the following parts would be numbered as 3.1., 3.2. and so forth instead of 4., 5. until the paragraph on current drugs (which is a different topic)

Response: Corrected.

In lines 308 to 310, the information is redundant and should be corrected. Also, in the following sentence, there is the impression that part of the sentence is missing. What did the authors want to state here?

Response: The sentences were corrected.

At last, in line 439, the authors state cancer stages 2-4. What exactly do they mean by that (WHO for CNS tumors? This would not be GBM then)? This should be explained a bit more thoroughly.

Response: Corrected. You are absolutely right - we changed the text on the «high-grade gliomas».

All changes in the article are highlighted in yellow to make it easier for the Editors to give a prompt decision on manuscript.

Yours faithfully, Dymova Maya.
